# Cultural Practices and Adoption of National Family Planning Communication Campaigns on Select Ethnic Groups in Nigeria

**DOI:** 10.3390/healthcare11040495

**Published:** 2023-02-08

**Authors:** Success Emmanuel Ojih, Babatunde Adeyeye, Ibe Ben Onoja, Evaristus Adesina, Funke Omole, Tolulope Kayode-Adedeji

**Affiliations:** 1Department of Mass Communication, Federal University Oye-Ekiti, Oye-Ekiti 371104, Nigeria; 2Department of Mass Communication, Covenant University, Ota 112104, Nigeria

**Keywords:** campaign messages, cultural practices, family planning, north-central Nigeria, National Family Planning Communication Campaigns (NFPCC)

## Abstract

This study evaluated the extent to which married Idoma (Benue State) and Igala people (Kogi State) in North-Central Nigeria were exposed to the 2017 National Family Planning Communication Campaigns. The study also examined their level of knowledge, the extent to which they adopted the campaign messages, and how Alekwu/Ibegwu and other socio-cultural factors influenced their level of adoption of the campaign messages. The study adopted a quantitative (questionnaire survey) research method. The data were subjected to a descriptive analysis, correlation, ANOVA, Pearson Product Movement Correlation (PPMC), and Binary Logistics Regression. The findings showed that the majority of the people were exposed to information on condoms, implants, and Intrauterine Contraceptive Devices (IUCDs) (Cuppar T) in the course of the campaign; however, most of them were not exposed to information on Oral Pills, Vasectomies, Tubal ligation and Injections. Findings also revealed that knowledge of modern family planning in the study areas (51.2%) was below the 85.8% national family planning knowledge threshold and far below the expected 95% target of the 2017–2020 family planning communication campaign goal. Findings equally showed poor adoption of the campaign messages due to their cultural beliefs. The study concluded that family planning was often accepted among people whose ways of life have been significantly altered in favour of the idea.

## 1. Introduction

Recently, discussions on surging population growth, migration, child and maternal health, infrastructural deficit, poverty, hunger, and malnutrition have generated increased concern amongst world leaders and international agencies [1]. The concentration of population growth in the poorest countries poses its challenges, making it more difficult to eliminate poverty and inequality, combat hunger and malnutrition, and expand educational and health systems, all of which are critical to the new sustainable development agenda’s success [2].

Various international conferences on population and development, reproductive health in general, and family planning in particular, such as the 1974 World Population Conference in Bucharest; the 1981 International Conference on Family Planning in Jakarta, Indonesia; the 1984 Mexico and 1994 Cairo International Conferences on Population and Development (ICPD); and the various Millennium Summits have consistently argued for deliberate actions and set a pact on population control across the globe, primarily through family planning. The call makes great sense in the African context, looking at the fact that the continuous growth of its population has propelled unhealthy competition for the limited available resources and social amenities such as health systems, schools, clean water supply, houses, electricity, waste management, to mention but a few [3]. Therefore, the discourse on family planning has significantly advanced in the existing literature [4,5,6,7,8,9,10,11].

To domesticate the idea of modern family planning, the Nigerian government has enacted different policies and programmes over the years. For instance, in 1988, the federal government promulgated its maiden Reproductive Health Policy, promoting the idea of one husband, one wife, and four children. This was later reviewed in 2004, giving rise to a similar commitment. Given this, government, media experts, and reproductive health officials have spent so much time creating the necessary awareness and promoting the desired behavioural change needed for the smooth implementation of family planning policies across the country. The 2017 National Family Planning Communication Campaign, with the motto “family planning services; safe and trustworthy,” aimed to reduce mother and new-born morbidity and mortality among Nigerian women of reproductive age (15–49 years) by promoting understanding and usage of modern contraceptives [12]. The initiative sought “7.3 million additional women of reproductive age in Nigeria who declared they do not want to get pregnant now or ever again” to use a contemporary Family Planning technique by December 2018, resulting in a 36 percent Contraceptive Prevalence Rate (CPR). However, many years down the lane and after the expiration of the year 2018, there still seems to be a low level of acceptance of family planning in the country, thereby creating the need for investigation into the possible impediment to the actualisation of the national family planning communication objectives despite the intervention efforts [6,13,14,15,16].

It is important to note that while only about 25.5% of women aged 15–49 years use any method of contraceptive and just 17.3% use modern contraceptives [12], the figure varies from region to region. Southern Nigeria tops the ranking with a prevalence rate of 36.8% in the southwest, 27.6% in the south–south and 18.7% in the southeast, compared to a prevalence rate of only 15.7% among women aged 15–49 years in the north-central part of the country where this study was conducted [12]. Within the north central, recent data suggest that Benue has a 17.1% modern contraceptive prevalence rate while Kogi has 18.3% which were slightly above the regional prevalent rate of 16.2% [12]. However, the same may not be said about Ogbadibo and Olamaboro Local Government Areas (LGAs) selected for this study from both states because of their similar conservative cultural practices and belief systems that set them apart from other Idomas and Igalas as well as other ethnic groups in both LGAs and states. 

The Idomas and Igalas in Ogbadibo and Olamaboro LGAs, respectively, have a unique and peculiar belief system that makes the use of modern contraceptives worth investigating. For instance, among these Idomas and Igalas, there is a belief that the use of modern contraceptives amounts to killing the unborn child. This belief is rooted in the teachings of their community deities called *Alekwu* and *Ibegwu*, respectively [17]. The principle of *Alekwu* and *Ibegwu* suggest that the killing of an unborn child through any means is an abomination that may attract the anger of their ancestors. To this end, the elders highly regulate the use of modern contraceptives. Although civilisation/westernisation has tremendously impacted the majority of African traditions and customs—the Idomas and Igalas inclusive, their belief system with regard to *Alekwu*/*Ibegwu* is an aspect that seems to be too powerful and consistent for any externally motivated repudiation or change that would be total.

An earlier investigation into the constrained agency in adopting family planning suggests that they are often multi-faceted and society specific [8]. Nevertheless, scholarship is often bereft of data that capture the influence of the socio-cultural dynamics of the varying ethnic groups on family planning and the reception and adoption of campaign messages on modern contraceptives, particularly in Central Nigeria.

Most family planning studies may not explicitly evaluate communication and or cultural aspect of the reproductive health issue as a significant variable. Still, scholars generally agree that the global realisation of the family planning target depends mainly on adequate communication and the promotion of a favourable cultural environment [18]. As a result of this implicit understanding, most family planning studies, especially in Africa, have made inferential extrapolations with behavioural change communication in mind. Such studies could be grouped into two major categories related to this current study. The first category includes research that primarily focused on the degree of recent contraceptive adoption and the agencies (factors) that influence that level of adoption. Indongo, for example, looked into the socioeconomic, demographic, and behavioural aspects that influence contraceptive use and method choice among young Namibian women, as well as measures to improve their access to health care and family planning services. The findings from the study implicitly demonstrate the need for communication and a re-evaluation of Namibia’s cultural values to enhance the attitude of the family planning service providers and adult population towards women’s use of contraceptives [19]. Similarly, Kiura investigated the constrained agency in family planning by *evaluating the perceptions, attitudes and experiences of Somali refugee women (in the Kakuma refugee camp in Kenya) on family planning* [20]. Kiura’s case study revealed low usage of contraceptives among the Somali women’s population due to cultural and religious practices, misinformation, illiteracy, and counterproductive approaches toward reproductive health in general [20]. The socio-cultural practice of the people was also identified as a decisive factor influencing the use of modern contraceptives among Somali women as seen in Kiura’s report [20]. This further lends credence to the assertion that cultural practice is a vital factor in family planning adoption in Africa and underpins the need for effective communication in achieving the desired family planning goals in the continent.

Additionally, Ndirangu et al. investigated the socio-economic and cultural challenges to family planning practices in Muranga North District [18]. Low use of family planning was also reported in the study location as a result of socio-economic and cultural factors such as bias, prejudices, and misconceptions; the value placed on children; gender inequality in the decision-making process regarding FP issues; and decreasing order, poor attitudes toward FP among most members of the community. As a result, the authors recommended that family planning programs should be intensified across the board to achieve better results. Therefore, the crux of this current study is to fill the identified research gap by evaluating the peoples’ level of exposure, knowledge, belief in Alekwu and Ibegwu, and other socio-cultural factors and their influence on the level of adoption of the 2017 National Family Planning Communication Campaign Messages. The following null hypothesis guided this study: there is no significant association between married people’s exposure to the campaign message on modern contraceptives and their level of adoption of the campaign message; there is no significant association between the married people’s level of knowledge and adoption of the campaign message on modern contraceptives in the study areas; there is no significant association between the people’s belief in Alekwu/Ibegwu and their level of adoption of the campaign message on modern contraceptives in the study area; there is no significant association between socio-cultural factors and the people’s level of adoption of the campaign message on modern contraceptives in the study area.

## 2. Materials and Methods

This paper used the survey research method as it enables the researcher to collect information from a representative sample of a target population and captures group dynamics among the various categories of respondents [21]. The research was conducted in Ogbadibo (Benue state) and Olamaboro (Kogi state) both in North Central Nigeria, where Alekwu and Ibegwu traditional customs reign supreme, respectively.

Using the Krejcie and Morgan sample size determination table, a sample size of 663 (six hundred and sixty-three) married men and women (or once married such as widowed, separated, divorced, or cohabiting) in Ogbadibo and Olamaboro LGAs of Benue and Kogi States, respectively, was selected for the study. The specific respondents for the survey were selected using multi-stage (probability and non-probability) sampling techniques to ensure proper representation because the population was large and complex.

The purposive sampling technique was used to select Ogbadibo (Benue state) and Olamaboro (Kogi state) *LGAs* because of their potency and resourcefulness to the research interest as earlier explained. Additionally, purposive sampling was repeated to select all the six districts—Imane, Ogugu, Okpo (In Olamaboro), Owukpa, Otukpa, and Orokam (In Ogbadibo)—in the selected LGAs owing to their inclusive relevance, to broaden the scope and since they all fall within the LGAs of the researchers’ interest. However, two wards were randomly selected to represent each district, making a total of 12 wards. Meanwhile, in each of the wards, the researcher purposively selected the most prominent village (the village with the highest population and physical infrastructure) to administer the instrument because they may be more enlightened and exposed to family planning messages or have more health service providers who could provide reliable and usable data for the study. In each of the Villages, two streets emerged from a simple randomisation process (balloting) involving all the major streets in each of the selected villages, which was considered for the choice of households where the questionnaire was purposively administered to only the married men and women (as earlier defined) since they fall within the ‘legitimate’ prescient of parenthood which is essential for a study of this nature.

The quota sampling technique ensured that both genders were fairly represented in the sample. The questionnaire was distributed on a 55% (363) to 45% (300) basis in favour of women. This was to reflect the general demographic dynamics in the areas, as seen in the 2006 population census result that stipulated that women were slightly in the majority in the study areas.

Meanwhile, within each street that emerged, a household consisting of one woman and one man that both satisfy the inclusive criteria of this research was sampled. This process was achieved using the Take-Pick Lottery Method of random sampling. YES or NO was written on pieces of paper and then appropriately folded, which were put in a container and thoroughly mixed at every stage of picking. A volunteer (a resident) in the household would then pick on its behalf. If the volunteer picks a ‘yes’, then a qualified man/woman (depending on availability) was sampled. In contrast, if a ‘no’ is picked, the researcher or his assistant moves on to the next household and repeats the same process. The following household was selected for replacement when there was a household that could not satisfy the inclusion criteria. If there is more than one qualified man and woman for selection, a simple random sampling was used to select the two respondents. This process was repeated until the required sample size was met based on the distribution quotas.

Eligible responders were allowed to understand the study’s goals and ask questions regarding the research and participants’ rights for ethical consideration. Each responder participated in this study voluntarily and was allowed to withdraw at any time without incurring any penalties. To ensure that the information they provided could not be traced, respondents were not obliged to reveal their names or traceable identities. As a result, respondents’ verbal agreement was gained. The research approach’s validity and reliability were determined using the pre-test reliability method and Cronbach’s alpha. Before the actual investigation, the results of the pilot poll suggested 85 percent validity. Statistical Package for Social Sciences was used to code the information gathered.

The questionnaire was structured into five sections. The first section had eight questions that sought to know the respondents’ demographic data: age, gender, marriage status, educational qualification, occupation, religion, duration of residence, and several children. The second section was made up of three questions that sought to establish the respondents’ level of exposure to the family planning campaign messages. The third section consisted of a Linkert table that has thirteen items that sought to know their level of knowledge on modern family planning methods as promoted by the campaign. Meanwhile, section D which had three Likert tables that contained sixteen items was designed to elicit responses on the extent to which the respondents adopted the campaign messages. The last section which had two close-ended questions and three Likert tables (with twenty-two items) sought to know how the *Alekwu* and *Ibegwu* cultural beliefs and practices enhance or impede on their level of reception and adoption of the 2017–2020 national family planning communication campaigns in the study areas.

The data were analysed using descriptive statistics (frequency count, percentage and mean), ANOVA, and inferential statistics (Pearson Product Moment Correlation and Binary Logistic Regression). 

All data obtained during the analysis were confidential and were used exclusively for this report. CHREC Protocol Assigned Number KSUTH/CMAC/ETHICAL/055/VOL.1/15 was obtained from the Kogi State University Teaching Hospital Research Ethical Committee for ethical consent.

## 3. Presentation of Results

A total of 663 respondents were sampled. Out of that, a total of 597 copies of the questionnaire (representing a 90% return rate) were retrieved and found useable, whereas 66 copies of the questionnaire representing 10%, were not retrieved or were retrieved but wrongly filled and, as such was not included in the analysis. Therefore, this study’s analysis, discussion and conclusion revolved around the 90% valid copies of the questionnaire.

Table 1 shows the data generated on the respondents’ demography and indicates that the majority of them (48%) were within their productive and sexually active age (15–45 years), 17% of them were at the brick of reproductive age and menopause (46–55 years), while only 35% of them were above 55 years when most women would have entered menopause. This suggests that the sampled age is the age at which the family planning campaign message was targeted.

According to the table, both sexes were represented almost equally (Male, 54% and female, 46%) even though more females than males were sampled. This is because the researchers recorded more invalid or wrongly filled copies of the questionnaire among the female respondents than the male respondents. However, the difference was not significant enough to affect the result negatively.

As seen in Table 1, the survey was dominated by: literate respondents (69.9%); artisans/traders and farmers (55.7%); Christians (79%); those who had resided in the study areas for 15 years or less (70%); still married and leaving together (76%); and those who had between 1–4 children (56%).

The various methods of modern contraceptives the respondents have been exposed to in the past two years were identified as seen in Table 2. The result showed that the respondents have majorly been exposed to only condoms, implants, and Intrauterine Contraceptive Device (IUCD) (Cuppar T) while only a few have been exposed to Oral Pill.

Looking at the general categorization of the respondents as also seen in Table 2 above, however, the majority of the respondents (55.1%) have been exposed to messages on modern contraceptives in the last two years while only 44.9% (268) of them have not been exposed to messages on modern family planning in the last two years. This therefore calls for increased awareness creation on the modern contraceptive methods the respondents have not been exposed to in the last two years.

Table 3 illustrates the weighted mean score of respondents’ sources of information on modern contraceptives and shows that the communication campaign has not achieved its aim of getting religious, community and traditional leaders to openly discuss family planning. It also suggests that the aim of getting spouses to discuss family planning in the selected local government areas in both states has not been fully achieved and neither has social media been adequately utilized to promote the knowledge of modern family planning in the study areas. The table showed that health workers or family planning service providers and the mass media (such as radio, television, newspapers, magazines, films, and movies) were the dominant sources of information on modern contraceptives among the people.

Table 4 below represents the weighted mean score and respondents’ categorization by their knowledge of modern contraceptives, as promoted in the 2017–2020 family planning campaign. The tables showed that knowledge of modern family planning in the study areas was high, but not significant enough to achieve the target of one of the campaign goals. This means that knowledge of modern family planning in the study areas (51.2%) was below the 85.8% national family planning knowledge threshold and far below the expected 95% target of the 2017–2020 national family planning communication campaign goal. This implies that the campaign’s objective of ‘increased’ knowledge was not achieved. The absence of basic information/knowledge in the study areas such as the ‘Green Dot’ logo signalling Family Planning services sites and Family Planning services and commodities being free in all public health facilities, etc., strongly suggests the campaign did not establish/achieve the desired effect, at least, on the studied locations.

Table 5 illustrates the percentage distribution and mean score on actions taken by the respondents as a result of their exposure to family planning campaigns as well as their categorization based on the actions taken. According to the tables, none of the actions expected of the respondents, as seen in the 2017–2020 national family planning campaign has been adopted. This means poor adoption of the family planning campaigns in the study areas.

Table 6 contains frequency and percentage distribution/weighted mean score of respondents’ responses on the influence of *Alekwu*/*Ibegwu* on the adoption of modern family planning as well as their categorization. According to the data in Table 6b, most of the respondents (71.4%, 426) believed that *Alekwu*/*Ibegwu* did not support the adoption of modern family planning. This suggests a negative influence of *Alekwu*/*Ibegwu* on the adoption of the 2017–2020 family planning campaign and calls for greater behavioural change communication.

Table 7 below represents the frequency and percentage distribution/weighted mean score of respondents’ responses on socio-cultural constraints to their adoption of modern family planning campaign as well as their categorization based on their socio-cultural constraints to the adoption of modern family planning campaigns. According to the data in Table 7 the majority of the respondents (52.4%, 313) were not constrained while only 47.6% (284) were constrained. However, frequency/percentage and mean analysis, as seen in Table 6a suggests that the respondent were constrained by *Alekwu*/*Ibegwu* as 50.3% (300) of them affirmed that the use of modern contraceptives will make married people incur the wrath of *Alekwu*/*Ibegwu* (1.5025 ± 0.50041).

Nevertheless, looking at the percentage of the respondents who were still constrained by the aforementioned factors, there is still a need to make modern contraceptives more readily affordable to the 40.4% of the respondents; design more comfortable modern contraceptives for 37.9% of them; establish more modern family planning service points to attend to the 37.45% of them who said none was close to them; intensify awareness campaign to correct the wrong impression among the 31.8% of the respondents that modern contraceptives could make them go barren as well as the 29.6% of them who did not have the necessary information on modern contraceptives to enable them to decide to use it or not.

As seen in Table 8 below, the null hypothesis that ‘there is no significant association between married people’s exposure to the campaign message and their level of adoption of the campaign message on modern contraceptives’ was rejected at 0.01 level of significance, meaning ‘there is a positive and significant association between married people’s exposure to the campaign message and their level of adoption of the campaign message on modern contraceptives. This implies that the more people get exposed to information on modern family planning, the more they adopt the content of the message.

Additionally, according to Table 8 above, the null hypothesis, which says that: ‘There is no significant association between the married people’s level of knowledge and adoption of the campaign message on modern contraceptives in the study areas’ was rejected at 0.01 level of significance as the test result indicates that there is a positive and significant association between the married people’s level of knowledge and adoption of the campaign message on modern contraceptives in the study areas. This suggests that the more people become knowledgeable on modern contraceptives, the more they adopt the family planning campaign messages on modern family planning and verse versa.

Lastly, according to the correlation analysis, as seen in Table 8 above, the null hypothesis states that: ‘There is no significant association between the people’s belief in Alekwu/Ibegwu and their level of adoption of the campaign message on modern contraceptives in the study area’ was rejected while. In contrast, the alternate hypothesis that: ‘There was a significant association between the people’s belief in Alekwu/Ibegwu and their level of adoption of the campaign message on modern contraceptives in the study area’ was upheld. This shows that the more the people believe in Alegwu/Ibegwu, the less they will adopt the campaign message on modern contraceptives.

As presented in Table 9, the coefficient of multiple determination (R^2^) of 0.55 was recorded; indicating that 55% variation in the adoption of modern contraceptive technologies in the study area is explained by the explanatory variables. According to the table, age (β = −0.060, *p*-value = 0.022 *), belief in the existence of Alekwu/Ibewu (β = −0.566, *p*- value = 0.013 *), and fear of Alekwu/Ibegwu punishment on offenders (β = −0.380, *p*-value = 0.020 *) have a negative and significant relationship with the adoption of the 2017–2020 family planning communication campaign messages in the study area; while marital status (β = 0.588, *p*-value = 0.027 *), spousal interaction (β = 0.343, *p*-value = 0.000 *) and information seeking behaviour (β = 0.268, *p*-value = 0.000 *) have a positive and significant relationship.

## 4. Discussion of Findings

The data analysed in this study were obtained from primary sources using the questionnaire. Exposure and knowledge of family planning, as well as the adoption of information on family planning, was generally low among the majority of the study population, just like in other climes in Africa, as reported by earlier research evidence [20]. Specifically, however, more people have been exposed to information on condoms, implants, and Intrauterine Contraceptive Devices (IUCD) (Cuppar T) within the last two years preceding this survey through Health workers or family planning service providers; Mass media (such as radio, television, newspapers, magazines, films, and movies); and Friends, relatives and community members in that order. Findings also show that most people in the study areas were not exposed to information on Oral pills, Vasectomy (preventing men from supplying sperm), Tubal ligation (Tying of fallopian tube), and Injection. In the same way, religious, community and traditional leaders, spouses and social media such as Facebook, WhatsApp, 2Go, Twitter, Instagram, etc., were not familiar sources of information on modern family planning to most people.

That most people were exposed to information on family planning through the mass media supports the earlier findings in other climes, as documented by [22,23,24]. According to [22], 71% of the respondents had exposure to family planning messages in the media (mobile phones, 48%; radio, 37%; and television, 29%) within the three preceding their survey. The popularity of the mass media as a source of information on family planning, as seen in the current study, may stem from the non-personal nature of mass media, especially since most men in the study areas found it more or less a taboo to discuss family planning with their spouses, or even the health officers and religious bodies such as the churches were indirectly prevented from discussing it openly as it was against their cultural beliefs.

However, this finding negates other findings by [25,26,27,28], where women were the dominant subject of study. According to them, even though channels of communication like broadcast media, print media, and interpersonal communication were used to promote family planning in the areas, interpersonal communication channels were more popular among women.

However, since exposure and knowledge were reported to have a positive relationship with the adoption of family planning among the select ethnic groups in Nigeria, there is a need to increase awareness campaigns on family planning using the popular media.

The study also showed that the people’s socio-cultural characteristics, such as age, sex, religion, occupation, and education level positively influence the adoption of the 2017–2020 family planning communication campaign messages in the study area. This finding supports [13] submission that the region of residence, gender, and socio-economic status of the population were significant predictors of contraceptive use in the country.

We equally reported that the people’s cultural beliefs and practices were constrained agency to the adoption of family planning in the study area as the more the belief in Alegwu/Ibegwu, the less the adoption of family planning in the study areas. That culture was a solid barrier to adopting the 2017–2020 national family planning communication campaign messages among the study population upheld earlier findings [16,18,19,20,23,29,30]. They also reported that culture stands out very prominently among the inhibiting factors in adopting family planning in other climes in Africa. Ndirangu et al. report poor adoption of family planning within the study location because of socio-economic and cultural factors such as bias, prejudices, and misconceptions; value attached to children; gender inequality in the decision-making process regarding FP issues; and poor attitudes regarding FP amongst most members of the community, in decreasing order [18,31,32]. They also opine that adults in Namibia and even the family planning health service providers ironically demonstrated a negative attitude towards young women’s use of contraceptives, which is a reflection of popular African cultural belief of the supposed sexual inactivity of single women in the continent. The cultural practice of the people was also identified as a decisive factor that influenced the use of modern contraceptives among Somali women, as seen in [20,33].

## 5. Conclusions

The use of modern contraceptives is often a function of the receptibility and adaptability of family planning promotional messages, as seen in the 2017–2020 national family planning communication campaign. However, research has shown how such messages are received and adopted as a function of several factors, including the level of exposure and cultural practices and beliefs. This means a society with a seemingly anti-family planning culture stood a greater risk of non-adoption of family planning campaign messages despite the proven benefits of FP.

The study revealed that the 2017–2020 national family planning communication campaign failed among Idoma and Igala ethnic groups in Ogbadibo and Olamaboro LGAs in Benue and Kogi states, respectively, due to their age barrier, belief in the existence of Alekwu/Ibewu, and fear of Alekwu/Ibegwu punishment on offenders. The study concludes that family planning will be accepted among the people only when their ways of life have been significantly altered in favour of the idea and calls for a more excellent state and community-specific educational programme that could help dispel the existing cultural beliefs and practices.

## Figures and Tables

**Table 1 healthcare-11-00495-t001:** Demographic Distribution of Respondents.

Demographic Variables	Frequency	Percentage
Age		
15–25 years	49	8
26–35 years	187	31
36–45 years	156	26
46–55 years	98	17
56–65 years	70	12
Above 65 years	37	6
Gender		
Male	322	54
Female	275	46
Educational Qualification		
No formal education	99	16.6
Primary	81	13.6
Secondary	266	44.6
Tertiary	151	25.3
Occupation		
Farming	152	25.5
Civil service	168	28.3
Artisan/Trading	180	30.2
Unemployed	95	16
Religion		
Christianity	470	79
Islam	88	15
African Traditional Religion	39	6
Years of Sojourn		
5 years and less	72	12
6–10 years	197	33
11–15 years	145	25
16–20 years	61	10
Above 20 years	121	20
Marital Status		
Married	452	76
Widowed	61	10
Cohabiting	6	1
Divorced	11	2
Separated	67	11
Number of Children		
None	63	10
1–2	129	21
3–4	207	35
5–6	118	20
7–8	58	10
9 and above	22	4
Total	597	100

**Table 2 healthcare-11-00495-t002:** Frequency and Percentage Distribution/Weighted Mean Score/Categorization of Respondents’ Exposure to messages on Modern Family Planning Method in the last Two Years.

	Always		Sometimes		Never		Mean	
Contraceptive Methods	Freq	%	Freq	%	Freq	%		Rank
Condom	390	65.3	133	22.3	74	12.4	2.5293	1st
Injection	102	17.1	212	35.5	283	47.4	1.6968	5th
Oral Pills	205	34.3	184	30.8	208	34.8	1.9950	4th
Implant	310	51.9	156	26.1	131	21.9	2.2998	2nd
IUCD	282	47.2	191	32	124	20.8	2.2647	3rd
Tubal ligation	95	15.9	174	29.1	328	54.9	1.6097	6th
Vasectomy	98	16.4	147	24.6	352	59	1.5745	7th
Overall Level of Awareness								
Respondents Awareness	Frequency		Percentage		Mean		St. Deviation	
7–13 (Not Heard of)	268		44.9		13.97		3.41	
14–21 (Heard of)	329		55.1					

**Table 3 healthcare-11-00495-t003:** Frequency and Percentage Distribution/Weighted Mean Score of Respondents’ Responses on their Sources of Information on Modern Family Planning.

Sources of Information	Yes		No		Mean	Remark
	Freq	%	Freq	%		
Health workers or family planning service providers	466	78.1	131	21.9	1.7806	Accepted
My religious, community and traditional leaders	223	37.4	374	62.6	1.3735	Rejected
Mass media (such as radio, television, newspapers, magazines, films, and movies)	444	74.4	153	25.6	1.7437	Accepted
Social media like Facebook, WhatsApp, 2Go, Twitter, Instagram, etc.	289	48.4	308	51.6	1.4841	Rejected
My spouse	260	43.6	337	56.4	1.4355	Rejected
Friends, relatives and community members	309	51.8	288	48.2	1.5176	Accepted

**Table 4 healthcare-11-00495-t004:** Frequency and Percentage/Mean/Categorization of Respondents’ Knowledge of Modern Contraceptives.

Knowledge on Modern Contraceptives	*n*	Yes		No		Mean	Std. Dev.	Remark
		Freq	%	Freq	%			
All Family Planning methods provided at public health facilities and private outlets are very safe, very effective and have been used by millions of women for years with no harmful effects, especially when used appropriately	597	356	59.6	241	40.4	1.5963	0.49105	Accepted
Once a woman has stopped using a modern Family Planning method, she will be able to get pregnant again if she wants to	597	343	57.5	253	42.4	1.5745	0.49483	Accepted
Every Family Planning method, like every medication, has side effects which can easily be managed and if there are major problems with a method, another Family Planning method can be used	597	332	55.6	265	44.4	1.5561	0.49726	Accepted
Women are advised to use a method that is appropriate for them	597	409	68.5	188	31.5	1.6851	0.46487	Accepted
Family Planning providers at public health facilities and private outlets are trained and are always ready to discuss all the women need to know about how to access Family Planning services.	597	337	56.4	260	43.6	1.5645	0.49624	Accepted
It is fairly easy to locate Family Planning service delivery points and outlets (where the new Family Planning logo is displayed)	597	290	48.6	307	51.4	1.4858	0.50022	Rejected
When it is a little difficult to locate Family Planning service delivery points and outlets, it is worth the effort to avoid an unplanned pregnancy.	597	342	57.3	255	42.7	1.5729	0.49508	Accepted
All religions approve of Family Planning methods and your religious leaders endorse it	597	144	24.1	453	75.9	1.2412	0.42817	Rejected
There is no problem discussing and jointly taking decisions on modern Family Planning methods to use with our marriage partners to delay the next birth or not to have any more and ensure a healthy family	597	291	48.7	306	51.3	1.4874	0.50026	Rejected
Approved Family Planning services sites are those health centres where the national ‘Green Dot’ logo is boldly pasted	597	276	46.2	321	53.8	1.4623	0.49900	Rejected
Family Planning services and commodities are free in all public health facilities.	597	249	41.7	348	58.3	1.4171	0.49349	Rejected
I can discuss Family Planning services with the nearest Service Providers so as to be well-informed about child spacing for the health and well-being of my family	597	353	59.1	244	40.9	1.5913	0.49201	Accepted
Health workers or family planning service providers are well equipped to provide information on the various contraceptive methods so that clients can make informed choices on the most appropriate methods for them	597	331	55.4	266	44.6	1.5544	0.49744	Accepted
Health workers or family planning service providers always use available Job Aids to provide quality Family Planning services to clients	597	306	51.3	291	48.7	1.5126	0.50026	Accepted
Overall level of Knowledge								
Respondents’ Knowledge	Frequency		Percentage		Mean		St. Deviation	
14–21 (Not Knowledgeable)	286		47.9		21.30		3.98	
22–28 (Knowledgeable)	311		52.1					

**Table 5 healthcare-11-00495-t005:** Frequency and Percentage Distribution/Weighted Mean Score/Categorization of Respondents’ Response to Actions taken as a result of their Exposure to Family Planning Campaign.

Adoption	Always		Sometimes		Never		Mean	Remark
	Freq	%	Freq	%	Freq	%		
I have sought correct and factual information about modern family planning methods from health facility or outlet that has “the Green Dot” logo in the last two year	159	26.5	182	30.5	257	43.0	1.8342	Rejected
I have used modern family planning methods to space my child in the last two years	114	19.1	156	26.1	327	54.8	1.6432	Rejected
I have used modern family planning methods to stop childbearing in the last two years.	115	19.3	92	15.4	390	65.3	1.5394	Rejected
I have discussed family planning with my spouse or partner in the last two years.	131	21.9	130	21.8	336	56.3	1.6566	Rejected
Overall Categorization by the Actions taken as a Result of their Exposure to Family Planning Campaign								
Action Taken	Frequency		Percentage		Mean		St. Deviation	
4–6 (Not Adopted)	327		54.8		6.67		2.655	
7–12 (Adopted)	270		45.2					

**Table 6 healthcare-11-00495-t006:** Frequency and Percentage Distribution/Weighted Mean Score/Categorization of Respondents’ Response on Influence of *Alekwu*/*Ibegwu* on Adoption of Modern Family Planning Methods.

Influence of *Alekwu*/*Ibegwu* on the Use of Contraceptives	Yes		No		Mean	Remark
	Freq	%	Freq	%		
*Alekwu*/*Ibegwu* supports any married person that wishes to use a Condom for family planning	180	30.2	417	69.8	1.3015	Rejected
*Alekwu*/*Ibegwu* supports any married person that wishes to use Intrauterine Contraceptive Device (IUCD) (Cuppar T) for family planning	82	13.7	515	86.3	1.1374	Rejected
*Alekwu*/*Ibegwu* supports any married person that wishes to use Inplant for family planning	118	19.8	479	80.2	1.1977	Rejected
*Alekwu*/*Ibegwu* supports any married person that wishes to use Oral Pills (tablet) for family planning	103	17.3	494	82.7	1.1725	Rejected
*Alekwu*/*Ibegwu* supports any married person that wishes to use Injection for family planning	115	19.3	482	80.7	1.1926	Rejected
*Alekwu*/*Ibegwu* supports any married person that wishes to use Tubal ligation (Tying of fallopian tube) for family planning	79	13.2	518	86.8	1.1323	Rejected
*Alekwu*/*Ibegwu* supports any married person that wishes to use Vasectomy (preventing men from supplying sperm) for family planning	90	15.1	507	84.9	1.1508	Rejected
Overall Categorization by Influence of *Alekwu*/*Ibegwu* on Adoption of Modern Family Planning Methods						
*Alekwu*/*Ibegwu* and Use of Contraceptives	Frequency		Percentage		Mean	St. Deviation
7–8 (Does not Support)	426		71.4		8.2848	2.09
9–14 (Supports)	171		28.6			

**Table 7 healthcare-11-00495-t007:** Frequency and Percentage Distribution/Weighted Mean Score/Categorization of Respondents’ Responses on Socio-Cultural Constraints to their Adoption of Modern Family Planning Campaign.

Constraints to Adoption of Modern Family Planning Campaign	Yes		No		Mean	Std. Dev.	Remark
	Freq	%	Freq	%			
It is expensive to buy modern contraceptives	241	40.4	356	59.6	1.4037	0.49105	Rejected
Modern contraceptives could make me go barren	190	31.8	407	68.2	1.3183	0.46619	Rejected
The use of modern contraceptives will make married people incur the wrath of *Alekwu*/*Ibegwu*	300	50.3	297	49.7	1.5025	0.50041	Accepted
There is no modern family planning service giver close to me	223	37.4	374	62.6	1.3735	0.48415	Rejected
I don’t have the necessary information on modern contraceptives to enable me to decide to use them or not	177	29.6	419	70.3	1.2970	0.45731	Rejected
Using modern contraceptives would make my spouse go for another partner	127	21.3	470	78.7	1.2127	0.40958	Rejected
Use of modern contraceptives will make people around me mock me	122	20.4	475	79.6	1.2044	0.40357	Rejected
I feel uncomfortable using any of the modern contraceptive methods	226	37.9	371	62.1	1.3786	0.48543	Rejected
Overall Categorization by Socio-cultural Constraints to the Adoption of Modern Family Planning Campaigns							
Constraints to Adoption	Frequency		Percentage		Mean		St. Deviation
8–10 (Not Constrained)	313		52.4		10.6879		2.28475
10–14 (Constrained)	284		47.6				

**Table 8 healthcare-11-00495-t008:** Correlation between Selected Variables and Adoption of the Campaign Messages.

Variable	PPMC	Significance	Decision
Exposure	0.391	0.000	Significant
Knowledge	0.323	0.000	Significant
Alekwu/Ibegwu	0.453	0.000	Significant

Correlation is significant at the 0.01 level (2-tailed).

**Table 9 healthcare-11-00495-t009:** Factors influencing the adoption of modern contraceptive technologies among respondents (*n* = 597).

	β	S.E.	Wald	*p*-Value	Decision
Age	−0.060	0.121	0.243	0.022 *	Sign.
Gender	0.165	0.237	0.487	0.485	Not Sign.
Education	0.027	0.085	0.103	0.748	Not Sign.
Years of Sojourn	−0.045	0.067	0.448	0.503	Not Sign.
Marital Status	0.588	0.266	4.877	0.027 *	Sign.
Number of Children	0.016	0.122	0.017	0.896	Not Sign.
Knowledge of Alekwu/Ibewu	0.003	0.038	0.005	0.942	Not Sign.
Awareness Alekwu/Ibewu	0.055	0.037	2.236	0.135	Not Sign.
Spousal interaction	0.343	0.082	17.398	0.000 *	Sign.
Information Seeking behaviour	0.268	0.037	51.458	0.000 *	Sign.
Believe in the existence of Alekwu/Ibewu	−0.566	0.229	6.108	0.013 *	Sign.
Fear Alekwu/Ibegwu of Punishment on offenders	−0.380	0.163	5.433	0.020 *	Sign.
Constant	−6.732	1.215	30.707	0.000	

Adjusted R-square (R^2^) = 0.55, * *p* < 0.05. Source: Field survey, 2021.

## Data Availability

Data for this study would be readily available through an email request to the corresponding author: babatunde.adeyeye@covenantuniversity.edu.ng.

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
