# Peer review of "Cultural Practices and Adoption of National Family Planning Communication Campaigns on Select Ethnic Groups in Nigeria"

_healthcare, 2023, doi:10.3390/healthcare11040495_

Round 1
Reviewer 1 Report
Paper 2042938 review to Healthcare – CULTURAL PRACTICES AND ADOPTION OF NATIONAL FAMILY PLANNING COMMUNICATION CAMPAIGNS ON SELECT ETHNIC GROUPS IN NIGERIA
All issues raised in this review can be considered to be minor reviews.
General considerations
The topic addressed is relevant, and this article is an asset to highlight methods that can be used to help with family planning and defining cultural behaviors. The article is well structured, the contents are well explained and articulated with each other. The literature review seems to be adequate, the data collection method and the analysis of the results obtained are presented in a perceptible way. All issues raised in this review can be considered to be minor reviews.
1. Structure
The structure of the article is well elaborated, but chapters are not numbered. Then, the authors must verify that this is in accordance with the journal's template, and if not, they may have to proceed with the referred numbering.
2. Title, Abstract and Keywords
· The title has key information and is appealing to readers.
· The abstract is well constructed. The main purpose of the study in presented, research methodologies are indicated and the main conclusions obtained are pointed out. Normally, the abstract should not contain acronyms, but only their full meaning. Acronyms should only be used in the text, and only when used for the first time should they be accompanied by their full meaning.
· The keywords are adequate.
3. Figures and tables
The figures and tables are all well numbered, and have good visual quality.
4. Grammar, spelling and syntax issues
The whole article it's well written in terms of grammar and spelling. But there were identified some aspects that should be improved/corrected, namely:
· Throughout the article there are several words and phrases in italics. Italics should be used when the sentence is written in another language and not for emphasis. Authors should check which form of emphasis is indicated by the journal template;
· In the 12th line of the 3rd paragraph of the Introduction - a space is missing after the reference [10]. It is suggest that the authors do a thorough analysis of the entire article looking for this type of error;
· In the 5th line of the 5th paragraph of the Introduction - the acronym FP must be accompanied by its full meaning, as this is the first time it is mentioned in the article;
· In the 5th line of the 1st paragraph of the Materials and Methods chapter - it is necessary to place a single quote sign (") that is paired with the one placed in the previous line.
5. Semantic and technical issues
The entire article is well explained. The issues are explained clearly and the concepts and ideas are well articulated between themselves. The data collection method is explained clearly and objectively. The qualitative and quantitative analyzes are presented in a perceptible way.
6. References
The list of references is well prepared, the number of references is appropriate to the depth of the theme's approach in the article. The references are strong in the scope of this investigation.
Reviewer 2 Report
This is a topical area of study that may turnout very good at the end if all suggestions are well accommodated to improve the manuscript.
Materials & Methods:
1. The content of the data is not explained i.e. the variables used whether they are continuous, categorical, etc.
2. Is there anything special about Ogbadibo and Olamaboro communities with respect to fertility rate or modern contraception use?
3. Did you use the 2002 population census sex composition distribution? Explain.
Results & Discussion
4. Your analysis is not robust enough to test your hypotheses. I suggest you run a multivariate model with adoption as the ultimate dependent variable, age, gender (add other background factors), exposure, knowledge, and belief as independent variables. Your first level of analysis is univariate (sample description of all variables), the second level is bivariate (correlations of predictors), and then the multivariate that I just described. I suggest you separate the results section from the discussion section because as it is, you have severely short-changed the discussion of results.
5. Your sample description including age, gender, and all other variables used in this study should be presented in a Table to make for easy comparison and save space. No need for the simplistic bogus pie charts on age and gender.
6. Your Table 1 is showing 'association' and not 'relationship.' There is a major difference between the two. Pearson correlation cannot test for the relationships stated in your hypotheses.
Conclusions
7. Your discussion and conclusions will change after you have accomplished the reanalysis suggested above.
8. Other comments are included in the body of the manuscript.

Reviewer 3 Report
The introduction it is recommended to strengthen, offering more information on the context of the country where the study is carried out, for example, on fertility rates in the country, contraceptive preferences of women of childbearing age, prevalence of use of methods, and methods more frequently used.
More information should be given about the implemented campaign and its dissemination strategies.
In the methods section, it is suggested to mention the main variables analyzed and how they were measured and operationalized. What was the approach used to establish cultural and economic barriers, biases and errors of interpretation and knowledge about contraceptive methods?
In the results section, it is recommended to present a table that summarizes the characteristics of the analyzed sample, according to whether or not the campaign advice was adopted. Likewise, present results of the variables of interest for men and women surveyed.
Findings that have not been presented in the results are discussed in the discussion section. This must be substantially strengthened.
Round 2
Reviewer 2 Report
1. I congratulate the authors for taking the time to go through the reviewers' comments and made necessary adjustments as necessary.
2. In general they should do one last edit and cut down on areas where they seem to be too verbose.
Author Response
Manuscript re-edited
Reviewer 3 Report
The authors have attended to the observations made previously. Intending to include relevant information that was not found in the first version under review, the current version presents an excessive number of tables that, in our opinion, can be reduced and the presentation of results made more efficient, through the combination of some of them, for example, the tables showing the correlations, could be grouped and presented in a single table.
Author Response
The effort of the reviewer is appreciated.
All corrections effected and highlighted in RED
Thank you.